# A Trend towards Diaphragmatic Muscle Waste after Invasive Mechanical Ventilation in Multiple Trauma Patients—What to Expect?

**DOI:** 10.3390/jcm12093338

**Published:** 2023-05-08

**Authors:** Liliana Mirea, Cristian Cobilinschi, Raluca Ungureanu, Ana-Maria Cotae, Raluca Darie, Radu Tincu, Oana Avram, Sorin Constantinescu, Costin Minoiu, Alexandru Baetu, Ioana Marina Grintescu

**Affiliations:** 1Department of Anesthesiology and Intensive Care, Clinical Emergency Hospital Bucharest, 014461 Bucharest, Romania; 2Department of Anesthesiology and Intensive Care II, Carol Davila University of Medicine and Pharmacy, 050474 Bucharest, Romania; 3Department of Clinical Toxicology, Carol Davila University of Medicine and Pharmacy, 050474 Bucharest, Romania; 4Department of Radiology, Carol Davila University of Medicine and Pharmacy, 050474 Bucharest, Romania; 5Department of Radiology, Victor Atanasiu National Aviation and Space Medicine Institute, 010825 Bucharest, Romania; 6Department of Radiology, Clinical Emergency Hospital Bucharest, 014461 Bucharest, Romania; 7Department of Anesthesiology and Intensive Care, Grigore Alexandrescu Clinical Emergency Hospital for Children, 011743 Bucharest, Romania

**Keywords:** multiple trauma, thoracic trauma, mechanical ventilation, diaphragmatic muscle, diaphragmatic dysfunction, ventilator-induced diaphragmatic dysfunction

## Abstract

Considering the prioritization of life-threatening injuries in trauma care, secondary dysfunctions such as ventilator-induced diaphragmatic dysfunction (VIDD) are often overlooked. VIDD is an entity induced by muscle inactivity during invasive mechanical ventilation, associated with a profound loss of diaphragm muscle mass. In order to assess the incidence of VIDD in polytrauma patients, we performed an observational, retrospective, longitudinal study that included 24 polytraumatized patients. All included patients were mechanically ventilated for at least 48 h and underwent two chest CT scans during their ICU stay. Diaphragmatic thickness was measured by two independent radiologists on coronal and axial images at the level of celiac plexus. The thickness of the diaphragm was significantly decreased on both the left and right sides (left side: −0.82 mm axial *p* = 0.034; −0.79 mm coronal *p* = 0.05; right side: −0.94 mm axial *p* = 0.016; −0.91 coronal *p* = 0.013). In addition, we obtained a positive correlation between the number of days of mechanical ventilation and the difference between the two measurements of the diaphragm thickness on both sides (r =0.5; *p* = 0.02). There was no statistically significant correlation between the body mass indexes on admission, the use of vitamin C or N-acetyl cysteine, and the differences in diaphragmatic thickness.

## 1. Introduction

Multiple trauma continues to be a global health problem, as it is so far the leading cause of death and disability [1]. Although real progress achieved through the development of advanced trauma life support principles, the morbidity associated with multiple trauma and mortality is still high [1,2].

Trauma-related respiratory failure may occur as a consequence of pulmonary contusion following blunt thoracic trauma but may also be induced indirectly by extrapulmonary factors such as associated traumatic brain injury, transfusions, fat embolism, and a systemic inflammatory response [3]. Acute respiratory distress syndrome (ARDS) after trauma may occur in more than 25% of cases with a variable onset depending on the severity of injuries [4]. Taking into account the multitude of risk factors for respiratory distress, mechanical ventilation is generally required for the management of multiple trauma patients [4].

The use of mechanical ventilation has significantly improved the outcome of multiple trauma patients through oxygenation and ventilation/perfusion ratio improvement [3,5]. However, inadequate mechanical ventilation proved to be more detrimental considering the high risk of ventilator-induced lung injury (VILI) [3]. Moreover, prolonged mechanical ventilation has an increased incidence of ventilator-associated pneumonia, a powerful determinant of increased mortality and prolonged hospital stay [6].

Recent data revealed that diaphragm muscle inactivity during mechanical ventilation might be associated with marked muscle atrophy and, subsequently, prolonged stay in the intensive care unit (ICU), weaning failure, and other unfavorable outcomes [7]. This new entity, called “ventilatory-induced diaphragmatic dysfunction (VIDD)”, may be characterized by the loss of both slow-twitch and fast-twitch fibers secondary to increased oxidative stress and exacerbated proteolysis [7,8].

Although VIDD diagnostic may be identified through a variety of imaging tools, very few research papers were dedicated to this topic. As a result, this study aims to evaluate diaphragm muscle dimensions in multiple trauma patients under mechanical ventilation using computed tomography (CT) scan.

## 2. Materials and Methods

A retrospective analysis of mechanically ventilated multiple trauma patients admitted to the Clinical Emergency Hospital of Bucharest was performed. The main goal of this study was to evaluate early changes in diaphragmatic thickness using CT-scan images ordered for different reasons during the patient’s stay in the ICU. Furthermore, the correlation between diaphragmatic measurements and the duration of mechanical ventilation was also determined.

All patients with multiple trauma, as defined by an Injury Severity Score ≥ 16, admitted into our hospital were included in the study group. Other inclusion criteria were mechanical ventilation for at least 48 h, as well as having performed two chest CT scan evaluations for different clinical reasons during their stay in the ICU. Multiple trauma patients with suspected or confirmed diaphragmatic rupture were not included in the final study group. Patients who had a history of invasive mechanical ventilation for more than 48 h in the last three months and comorbidities such as chronic obstructive, pulmonary disease (COPD), neoplasia, severe undernutrition (Body Mass Index (BMI) < 18 kg/m^2^), autoimmune (e.g., polymyositis, dermatomyositis, systemic sclerosis) or neurological diseases (e.g., multiple sclerosis, myasthenia gravis) were excluded. Patients under long-term use of glucocorticoids or anabolic hormones were also excluded from the final study group. According to the local protocol and the latest guideline of the European Society for Clinical Nutrition and Metabolism (ESPEN), all patients included in the final study group benefited from enteral nutritional support in the first 48 h [9]. Calorie intake was based on an estimated energy expenditure measured through indirect calorimetry, and a protein intake of 1.3 g/kg body weight was targeted.

Diaphragmatic thickness measurements were performed twice by two independent radiologists using the CT scan images obtained on admission and after day 5. The celiac axis was used as a reference point. At this level, diaphragm muscle thickness was measured on axial as well as coronal images. Mean obtained values and differences between the two sets of measurements were used for the final analysis.

Demographic data and clinical and laboratory parameters, Trauma scores (e.g., Injury severity score—ISS), and ICU predictive scores (e.g., Acute Physiology and Chronic Health Evaluation II—APACHE II) were collected from electronic patient records and analyzed.

The whole study protocol was designed according to STROBE guidelines and was approved by the Institutional Ethics Committee of the Clinical Emergency Hospital in Bucharest.

### 2.1. Statistical Analysis

Statistical analysis of the database was performed using MeDCalc 14.1. A Bland—Altman concordance analysis between the two sets of measurements made by different radiologists was performed, and a simple linear regression was adjusted for the time interval between the two CT scans. In the linear regression analysis, the F-test derived from the ratio of the mean square regression and the mean square residual was used to evaluate whether the variability of the regression model could be explained by variations in the dependent variable or could be attributed to random chance. In other words, a significant F-test validated the relationship between the independent and the dependent variables.

The Spearman correlation coefficient (r) was also measured in order to describe the relations between the diaphragmatic thickness and other variables. Knowing that the correlation coefficient r ranges between −1 and 1, a negative r value indicates an inversely proportional relationship between variables, while a positive correlation confirms a proportional relation. When the r value is null, no correlation is validated. *p* value less than 0.05 was considered significant.

#### Sample Size

In order to establish the correlation between diaphragmatic thickness and the duration of mechanical ventilation, the sample size was estimated using the sample size calculator for correlation coefficients. As a result, 23 patients were needed in order to achieve a study power of 80% (type 1 error alpha with significance level 0.05 and type 2 error beta 0.2) with an estimated correlation coefficient of 0.55.

## 3. Results

Between 2019 and 2020, 105 multiple trauma patients were admitted into the Intensive Care Unit. Only 63 (66.15%) patients were admitted into our unit per primam, and 42 (44.1%) patients were transferred from other hospitals after variable intervals. After considering the exclusion criteria, the final study group included 24 patients. Details regarding excluded patients are presented in Figure 1.

The median age in the study group was 59 years (16–81) (Figure 2), and 15 patients (62.5%) were male. Nutritional status evaluation based on BMI calculation indicated a median value of 26 kg/m^2^ (Figure 2).

Trauma severity assessment revealed a median ISS score of 35 (18–57) (Figure 3). A detailed traumatic assessment is presented in Table 1. Disease severity assessment on ICU admission indicated a median APACHE II score of 19 (9–57) (Figure 3) and a median Sequential Organ Failure Assessment (SOFA) score of 7.

The mean duration of mechanical ventilation was 16.27 ± 9.1 days (median 16.5, range 28) (Figure 4), and the mean ICU stay was 26 days ± 12 days. Between the CT exams, all patients received only pressure-controlled mechanical ventilation with a mean PEEP value of 6 ± 1.1 cm H_2_O. While admitted into the ICU, 18 patients (75%) received therapy with corticosteroids (the equivalent of dexamethasone 8 mg/day for at least 5 days) for different medical reasons, and 11 patients (45.8%) had their treatment supplemented with intravenous vitamin C (750 mg/daily). Antioxidant therapy with N-acetylcysteine (900 mg/daily for at least 5 days) was prescribed for only 6 patients (25%). No patient received muscle relaxants during the analyzed period. Out of the final study group, 18 patients (75%) were successfully extubated, and tracheostomy was necessary for 3 patients (7.2%).

To identify systematic biases or trends in the differences between the two radiological measurements, we performed a Bland—Altman concordance analysis (Figure 5). To calculate the measurement biases, the formula A − B/Average was used. The resulting measurement biases obtained were small (−0.02 axial left, 0.05 axial right, 0.08 coronal left, and −0.01 coronal right), and the 95% interval between the limits of agreement concludes that the two main investigators did not have large measurement differences and did not distort the resulted data.

All patients underwent at least two chest CT scan evaluations. The mean interval between the two radiological investigations was 6.08 ± 5.8 days. Considering that the second set of measurements could not be performed on exactly the same day for every patient from the study group, a regression analysis adjusted for the interval between the two radiological investigations was performed (Figure 6).

Time is represented on the OX axis. The difference (in millimeters) between the axial and coronal measurements from the two CT scans is shown on the OY axis. The slope of the line corresponding to axial differences has a value of 0.16, with a Y-intercept of −0.43 and an X-intercept of 2.62. The slope of the coronal line has a value of 0.13 with a Y-intercept of −0.38 and an X-intercept of 2.76. The coronal slope (F = 6.18, *p* = 0.017) differs significantly from the value of 0 in statistical terms. The axial slope (F = 7.52, *p* = 0.094) did not meet the significance threshold. Based on these statistical data, a difference of 0.16 mm can be observed each day throughout the duration of mechanical ventilation procedures.

A proportional relationship was found between the diaphragmatic measurements obtained both on axial sections (left: r = 0.48, *p* = 0.035; right: r = 0.46, *p* = 0.046) and coronal sections (left: r = 0.62, *p*= 0.004; right: r = 0.46, *p* = 0.04), on the one hand, and the duration of mechanical ventilation (Figure 7), on the other hand, suggesting that prolonged mechanical ventilation is directly associated with diaphragmatic thinning.

## 4. Discussion

The current study demonstrates that multiple trauma patients undergoing mechanical ventilation for more than 48 h may develop diaphragmatic dysfunction, characterized by a significant decrease in muscle thickness and a prolonged duration of mechanical ventilation.

Over the last decade, the management of multiple trauma patients benefited from continuous advances obtained in the fields of trauma life support or damage control resuscitation. However, less priority has been given to the impact of all implemented supportive measures [1,10]. It has already been proven that the occurrence of respiratory dysfunction after a traumatic injury and the need for invasive mechanical ventilation may be independent factors responsible for worsening the long-term outcome [4]. Trauma-related respiratory distress may be induced by a direct lung injury (e.g., chest trauma, aspiration of gastric content), as well as by the systemic inflammatory response secondary to traumatic shock or by extensive transfusions [11,12].

Although mechanical ventilation is undoubtedly a lifesaving therapy for many multiple trauma patients, excessive stress and strain applied to an injured lung during mechanical ventilation may cause an antithetical effect [13,14]. The use of elevated tidal volumes and driving pressures, as well as the inappropriate positive end-expiratory pressure values, are associated with exacerbated inflammatory responses (biotrauma) that generate additional detrimental effects on traumatic lung injuries [14,15].

In addition to VILI, secondary effects of mechanical ventilation on diaphragmatic muscle were also reported [8]. For patients undergoing invasive mechanical ventilation, several mechanisms of diaphragmatic myotrauma were described, such as excessive ventilatory support (over-assistance), inadequate diaphragmatic work unload (under-assistance), eccentric diaphragm contractions during patient-ventilator asynchrony or longitudinal atrophy caused by increased PEEP values [16]. All these mechanisms are finally translated into exacerbated inflammation, mitochondrial dysfunction, oxidative stress, autophagy, and protein catabolism [7,17]. Whether mechanical ventilation-related autophagy has a real detrimental effect is something that remains unknown, given that this process promotes the clearance of altered mitochondria and muscle function improvement [7]. Nevertheless, recent data suggest that calpain, a skeletal muscle protease, plays a central role in VIDD through the activation of the ubiquitin–proteasome system or caspase-3 [17].

Several diagnostic tools are proposed for diaphragmatic muscle dysfunction. However, in multiple trauma patients diaphragmatic, CT evaluation may be more advantageous, considering that it may provide data regarding diaphragmatic structural integrity and subdiaphragmatic processes that may influence respiratory mechanics [18,19]. Taking into account that CT scan has become a routine diagnostic tool, using the already acquired images for diaphragm muscle composition evaluation may be very time- and cost-efficient [20]. For all patients included in this current retrospective analysis, CT images ordered for different medical reasons were used.

In our study group, multiple trauma patients undergoing mechanical ventilation suffered a decrease in diaphragmatic muscle thickness after a relatively short duration of mechanical ventilation (6.08 days). Lee et al. reported that changes in diaphragmatic thickness were identified on CT scan examinations after a mean period of 18 days [21]. However, diaphragmatic ultrasound evaluation revealed that the thinning of the muscle might be detected even earlier, after only 48 h of mechanical ventilation [22]. A recent study by Gatti et al., who evaluated the thickness of the diaphragmatic muscle in six different areas, also revealed that the thickness of the left posterior pillar decreases with mechanical ventilation duration [23]. Moreover, it has also been demonstrated that diaphragm thickness correlates with the skeletal muscle index in patients undergoing mechanical ventilation, including multiple trauma patients [23,24].

Assuming that in the final study group, only multiple trauma patients were included, without any potential muscular dysfunction, the rate of the decrease of diaphragmatic thickness was remarkably high in comparison with similar research data.

### Limitations

The main limitation of this study is the retrospective data evaluation. Considering that diaphragmatic dysfunction was evaluated only through a retrospective analysis of CT scans, no functional imaging was available.

Despite achieving a calculated sample size, one of the main limitations of this study is the limited sample size (n = 24).

At the moment, there is no recommended “reference” point for diaphragmatic measurements. In this context, the celiac axis was used as a reference point on both axial and coronal images based on previously published data [21]. However, recent data suggest that multiple reference points should be used considering the heterogeneous structure of the diaphragm [23].

Taking into account that the skeletal muscle index is considered an independent risk factor for prolonged mechanical ventilation, our current research lacks further data regarding body composition.

Further research may be needed in order to evaluate anatomical and functional diaphragmatic changes by combining CT scanning and ultrasonography.

## 5. Conclusions

Our current research suggests that diaphragmatic morphological changes may occur surprisingly faster after a relatively short duration of invasive mechanical ventilation in patients without any prior evidence of chronic comorbidities.

Evaluation of diaphragmatic dysfunction may be performed with a variety of imagistic tools. Computed tomography examination, routinely used for primary and secondary evaluation of multiple trauma patients, may also offer the advantage of diaphragmatic evaluation.

## Figures and Tables

**Figure 1 jcm-12-03338-f001:**
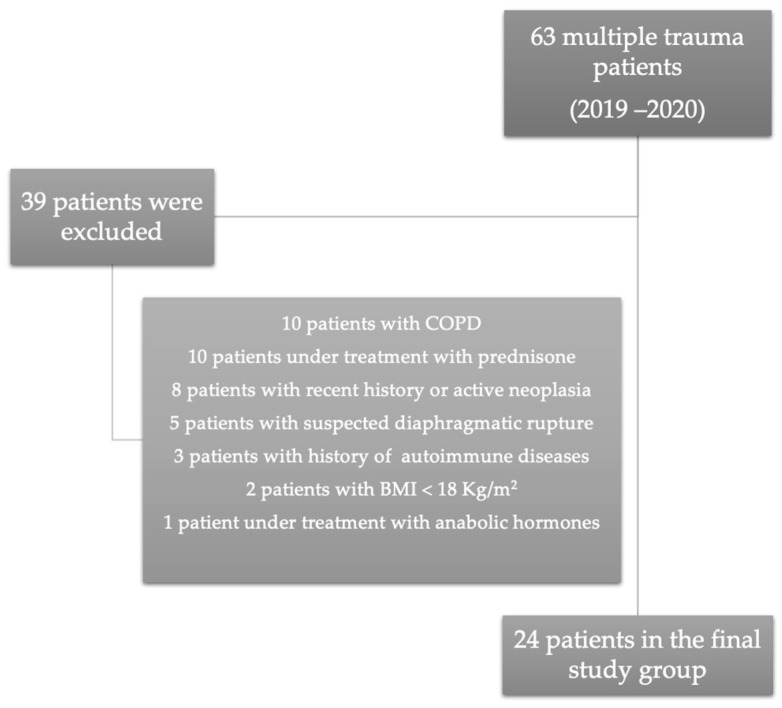
Final study group after the application of exclusion criteria.

**Figure 2 jcm-12-03338-f002:**
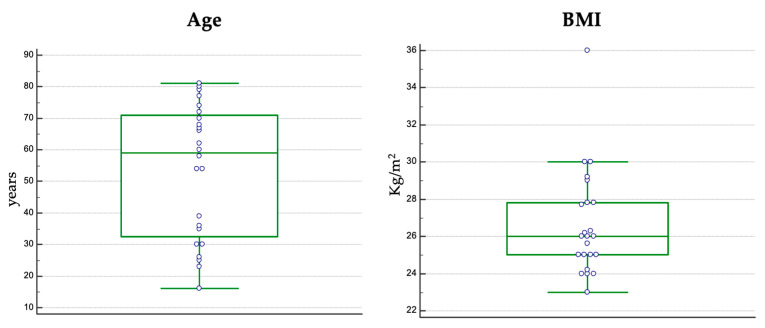
Age and BMI distribution in the final study group.

**Figure 3 jcm-12-03338-f003:**
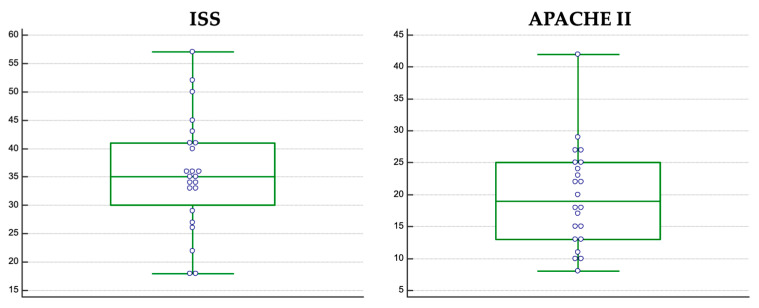
ISS and APACHE II scores distribution in the final study group.

**Figure 4 jcm-12-03338-f004:**
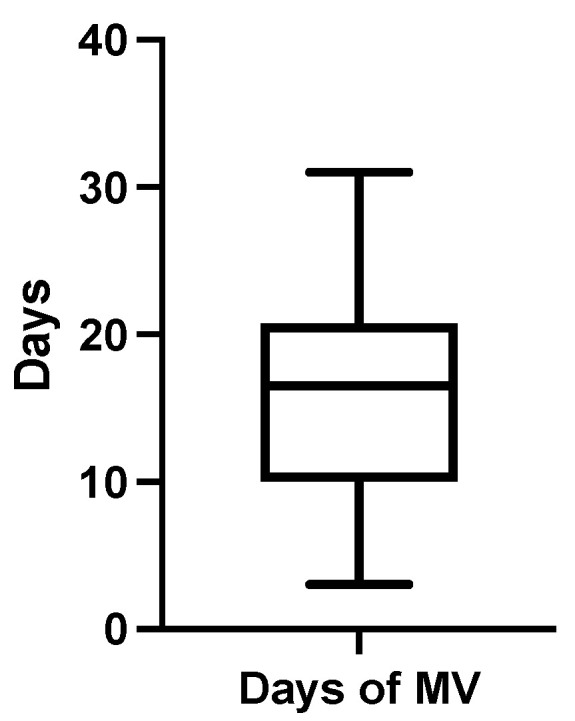
Duration of mechanical ventilation distribution.

**Figure 5 jcm-12-03338-f005:**
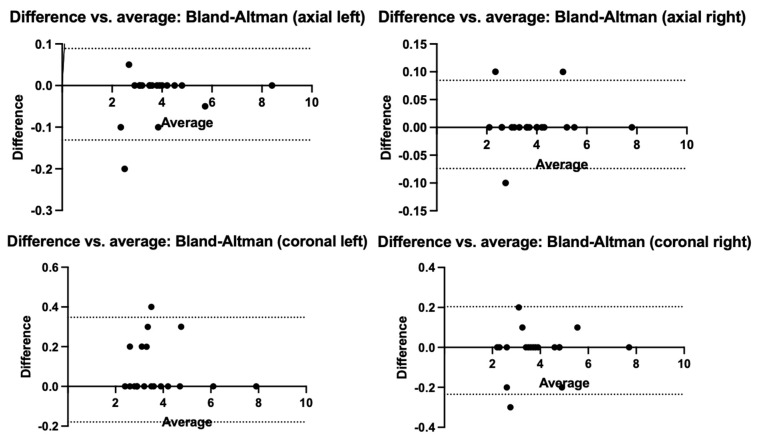
Bland—Altman concordance analysis for the two sets of measurements performed on admission.

**Figure 6 jcm-12-03338-f006:**
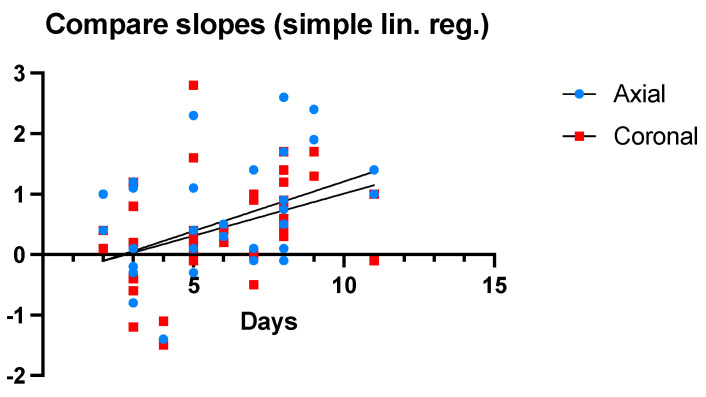
Simple linear regression adjusted for the time interval between the two CT scans.

**Figure 7 jcm-12-03338-f007:**
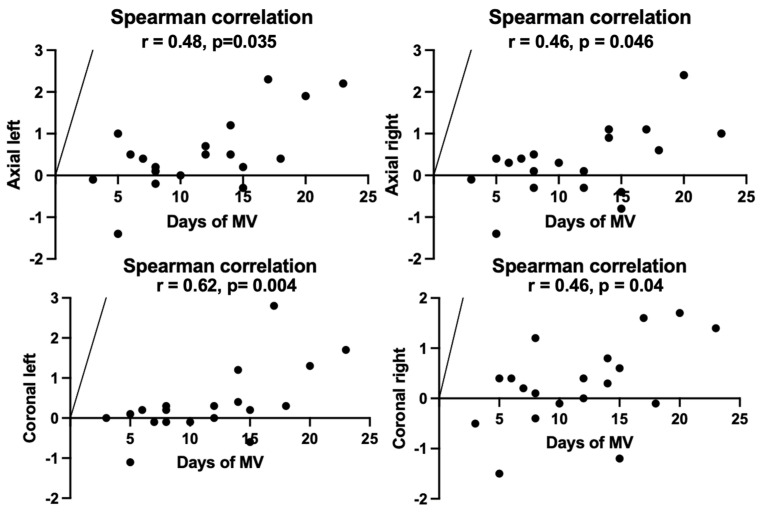
Correlation between the differences in the two diaphragmatic measurements and the duration of mechanical ventilation.

**Table 1 jcm-12-03338-t001:** Traumatic assessment in the study group.

Body System	Number of Patients
Trauma brain injury	19 (79.1%)
Facial injury	9 (37.5%)
Thoracic trauma	22 (91.6%)
Abdominal trauma	7 (29.1%)
Pelvic injury	9 (37.5%)
Extremity injuries	17 (70.8%)

## Data Availability

All presented data are available on demand.

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
