# Peer review of "A Trend towards Diaphragmatic Muscle Waste after Invasive Mechanical Ventilation in Multiple Trauma Patients—What to Expect?"

_jcm, 2023, doi:10.3390/jcm12093338_

Round 1

Reviewer 1 Report

Dear Dr. Mirea and co-authors,

Thank you for sending this interesting manuscript for review.
Please note the comments relating to the manuscript:

1. Title: The title does not adequately reflects the manuscript content. The study design is unclear from the title. Kindly add that to the title. Also, it is a "trend toward" diaphragmatic muscle wasting.

2. Abstract: Adequate.

3. Methodology:

"polytraumatized" - kindly de-verbize the noun.

Page 2, line 74: "primarily admitted to our hospital" - it is unclear by this statement if there was a secondary group or anyone not admitted to the primary site.

Page 2, Line 82: "patients with under treatment with corticosteroids or anabolic hormones" - unclear what under treatment with corticosteroids means.

Page 2,  Line 86: Nutritional support was progressively initiated after 48 hours, for all patients included in 84 final study group, according to the latest European Society for Clinical Nutrition and Metabolism guideline - again, a very vague statement as this is a retrospective study.

Results:

By eliminating the spread of the diaphragmatic thickness and their overlap from results, the data misrepresents. There is a significant overlap. The authors should acknowledge that in the results section. Any attempt to not do that would be misleading to the reader.

The second issue with the results is that the sample size has not been calculated. This leads us to the issue of the study being underpowered/ inadequately powered. This is a major limitation and should be acknowledged.

Discussion: 

The discussion regarding excessive PEEP, biotrauma, or direct mechanical trauma is irrelevant to the case. This is because there is no relevant population in this study.

"Assuming that in the final study group were included only multiple trauma patients 198 without any potential muscular disfunction, rate of the decrease in diaphragmatic thickness was remarkably high. " - That is a difficult assumption to make, considering the well-established role of critical care myopathy, and no adjustment for factors that cause it.

Conclusion:

The first 3 sentences can be omitted.

Page 7, Line 213: "with no previous history of systemic pathologies" - authors did not account for all relevant systemic pathologies.

I wish you good luck with your manuscript

Author Response

Thank you very much for all your recommendation. We are very grateful for the substantial improvement of our manuscript.

1. Title: The title does not adequately reflects the manuscript content. The study design is unclear from the title. Kindly add that to the title. Also, it is a "trend toward" diaphragmatic muscle wasting.

Title changed “A trend towards early diaphragmatic muscle waste after invasive mechanical ventilation in multiple trauma patients”

3. Methodology:

"polytraumatized" - kindly de-verbize the noun.

polytraumatized was replaced with Multiple trauma

Page 2, line 74: "primarily admitted to our hospital" - it is unclear by this statement if there was a secondary group or anyone not admitted to the primary site.

Primarily was excluded.

Page 2, Line 82: "patients with under treatment with corticosteroids or anabolic hormones" - unclear what under treatment with corticosteroids means.

"patients with under treatment with corticosteroids was rephrased   - patients under patients under long-term use of glucocorticoids

Page 2,  Line 86: Nutritional support was progressively initiated after 48 hours, for all patients included in 84 final study group, according to the latest European Society for Clinical Nutrition and Metabolism guideline - again, a very vague statement as this is a retrospective study.

A more clear paragraph dedicated to the nutritional protocol was added: According to the local protocol and to the latest European Society for Clinical Nutrition and Metabolism (ESPEN) guideline, all patients included in the final study group benefited from enteral nutritional support in the first 48 hours[9]. Calorie intake was based on estimated energy expenditure measured through indirect calorimetry and protein intake of 1.3 g/Kg body weight was targeted.

Results:

By eliminating the spread of the diaphragmatic thickness and their overlap from results, the data misrepresents. There is a significant overlap. The authors should acknowledge that in the results section. Any attempt to not do that would be misleading to the reader.

"We apologize, but we are having difficulty understanding. Could you please provide further clarification?" 

The second issue with the results is that the sample size has not been calculated. This leads us to the issue of the study being underpowered/ inadequately powered. This is a major limitation and should be acknowledged.

Considering that the primary outcome of the study was to establish the correlation between diaphragmatic thickness and the duration of mechanical ventilation, sample size was estimated using the sample size calculator for correlation coefficient. As a re-sult, 23 patients were needed in order to achieve a study power of 80% (type 1 error al-pha with significance level 0.05 and type 2 error beta 0.2) with an estimated input of detection correlation coefficient of 0.55.

Discussion: 

The discussion regarding excessive PEEP, biotrauma, or direct mechanical trauma is irrelevant to the case. This is because there is no relevant population in this study.

The reason for these explanations was that excessive PEEP, biotrauma, or direct mechanical trauma are already independent risk factors for ventilatory-induced diaphragmatic dysfunction for every critically ill patients , including multiple trauma patients.

"Assuming that in the final study group were included only multiple trauma patients 198 without any potential muscular disfunction, rate of the decrease in diaphragmatic thickness was remarkably high. " - That is a difficult assumption to make, considering the well-established role of critical care myopathy, and no adjustment for factors that cause it.

Rephrased as : Assuming  that in the final study group were included only multiple trauma patients without any potential muscular disfunction, rate of the decrease in diaphragmatic thickness was remarkably high, in comparison with similar research data

Conclusion:

The first 3 sentences can be omitted.

Deleted

Page 7, Line 213: "with no previous history of systemic pathologies" - authors did not account for all relevant systemic pathologies

Rephrased as : without any prior evidence of chronic comorbidities

Reviewer 2 Report

I've read the paper with great interest but It should be improved  in some points (major revisions). Probably It should be necessary ask to an statistical consultant.

In order to improve the paper an Bland-Altman concordance analysis between the two radiologist should be done. 

Line 121: Mean duration of mechanical ventilation should be reported also with median and range. It's presumable that the distribution is not symmetric. Furthermore It's preferable include a boxplot with this distribution.

Line 99: A Paired T test or T test for two independent groups analysis was adopted? The latter is the appropriate analysis. Please specify.

Line 99: it is non clear if all 24 patient enrolled in the study was measure exactly at 5 days. In fact subsequently in line 130 It is reported that the mean interval between the two radiological investigation was 6.08 +/- 5.8 days. If the difference between the two radiological investigastion is different (in term of days) between the 24 patients T test analysis not adjust the comparison for this latter difference. This is an important bias. An alternative could be a regression analysis adjusted for the interval between the two radiological investigations. Please, apply the correct analysis and specify it in the Statistical Analysis and Material and Methods paragraphs.

Figure 6: Days of MV correspond to the days of measurement for each patient? Why in left side there are 3 measurement and in right side there are two measurement at day 10?

Line 100: Figure 6 shows a non linear association, so Spearman correlation should be more appropriate than Pearson Correlation.

It's auspicable improve the discussion exploring moreover the non linear trend of the reduction of the Diaphragmatic muscle. For example, form the Figures 6 this reduction is very important from about day 12 and at the contrary it appear irrelevant before that day. It should be done also an explorative (with nominally p-value) segmented (piecewise) regression analysis that explore the rate of reduction before and after the day 12.

It could be interesting to ad a comparative explorative analysis (with nominally p-value) between the principle traumatic assessment area.

Finally a Statistical Analysis paragraph should be created and some items from Material and Methods should be moved in this new section

Author Response

Thank you very much for all your suggestions. We are really grateful, considering that the results are now enriched.

I have made the changes as follows:

In order to improve the paper an Bland-Altman concordance analysis between the two radiologist should be done. 

A Bland-Altman concordance analysis was added in the manuscript.

Line 121: Mean duration of mechanical ventilation should be reported also with median and range. It's presumable that the distribution is not symmetric. Furthermore It's preferable include a boxplot with this distribution.

Median and range were added and a boxplot expressing this distribution was also included.

Line 99: A Paired T test or T test for two independent groups analysis was adopted? The latter is the appropriate analysis. Please specify.

T test for two independent groups was specified

Line 99: it is non clear if all 24 patient enrolled in the study was measure exactly at 5 days. In fact subsequently in line 130 It is reported that the mean interval between the two radiological investigation was 6.08 +/- 5.8 days. If the difference between the two radiological investigastion is different (in term of days) between the 24 patients T test analysis not adjust the comparison for this latter difference. This is an important bias. An alternative could be a regression analysis adjusted for the interval between the two radiological investigations. Please, apply the correct analysis and specify it in the Statistical Analysis and Material and Methods paragraphs.

Simple linear regression adjusted for the time interval between the two CT scans was added in the results section

Figure 6: Days of MV correspond to the days of measurement for each patient? Why in left side there are 3 measurement and in right side there are two measurement at day 10?

We think that it is not applicable anymore since a Spearman correlation test was applied, and the figure was changed.

Line 100: Figure 6 shows a non linear association, so Spearman correlation should be more appropriate than Pearson Correlation.

A Spearman correlation test was applied.  

It's auspicable improve the discussion exploring moreover the non linear trend of the reduction of the Diaphragmatic muscle. For example, form the Figures 6 this reduction is very important from about day 12 and at the contrary it appear irrelevant before that day. It should be done also an explorative (with nominally p-value) segmented (piecewise) regression analysis that explore the rate of reduction before and after the day 12.

Not applicable anymore since A Spearman correlation test was applied and the new graph has a new apperance.

It could be interesting to ad a comparative explorative analysis (with nominally p-value) between the principle traumatic assessment area

Although we  agree that this analysis would be a valuable addition for our paper, we had to consider that the traumatic assessment in the study group is relatively homogenous. For exemple from 22 patients with thoracic trauma -19 had also TBI.

Finally a Statistical Analysis paragraph should be created and some items from Material and Methods should be moved in this new section

A new section dedicated to the statistical analysis was done.

Reviewer 3 Report

Thank you for the opportunity to review this original article by Mirea and colleagues. The topic chosen by the authors is very interesting, since the ventilatory – induced diaphragmatic dysfunction (VIDD) has been widely investigated in recent years and it has been associated with clinical outcome in critically ill patients.

The text is concise and well written. I would just like to add a few comments that may help improve the utility of this data.

1)     Several diagnostic tools have been proposed to evaluated diaphragmatic muscle dysfunction and particularly diaphragm thickness. CT scanning has several disadvantages (the need to transfer the patient, X-ray exposure, lower spatial resolution and contrast with surrounding structures). The possibility to assess several portions of the diaphragm is the main advantage of CT scanning. The authors measured diaphragm thickness in a single point using the celiac axis as the reference both on axial and coronal images. How did the author evaluate that the diaphragm thickness measured at the celiac axis is representative of the whole diaphragm? A recent manuscript by Gatti and colleagues (doi:10.3390/diagnostics12112890) evaluated diaphragm thickness at different positions and it is reported a different characterization of the diaphragm based on the level of diaphragm assessment at the CT evaluation. I would recommend to reference this manuscript and comment your findings in light of this recent study.

2)     Recent studies reported a correlation between parameters of body composition and outcomes in critically ill patients. An independent association between low skeletal muscle index (SMI) and high myosteatosis (MS) with the 90 – day mortality has been recently demonstrated by Giani et al. (doi: 10.1016/j.nut.2022.111687)confirming the already demonstrated relationship between a low SMI and mortality in different ICU population, included trauma patients. Further, the recent study by Gatti et al. (doi:10.3390/diagnostics12112890) described that SMI value was inversely correlated with the duration of mechanical ventilation before the CT scanning, suggesting that the muscle mass might decrease over prolonged days of mechanical ventilation. Based on the studies mentioned above, the evaluation of SMI on ICU admission ad after day 5 should be investigated since a baseline sarcopenic condition of the patient may play a role on the duration of mechanical ventilation. Please comment on this and reference your findings considering these novel studied recently reported in the literature. These points are crucial. Can the authors consider to obtain data on body composition indexes at the patient admission as they have computed tomography images?

3)     Assisted mechanical ventilation modes, such as pressure support ventilation, can lead to partial restoration of diaphragm thickness. In the time between the admission CT scanning and the second imaging, have all patients been ventilated using controlled mechanical ventilation? This is a crucial aspect that the authors should clearly report.

4)     Literature is not univocal about the influence of the neuromuscular blocking drugs (NMBDs) in the pathogenesis of the diaphragm atrophy. Were all patients in your population treated with neuromuscular blocking agents? Did the authors find any association between diaphragmatic thinning and the NMBDs days?

Author Response

Thank you for your precious indications. I made all the suggested changes as follows:

1)     References was added as well as  - these two paragraphs 

A recent study by Gatti et al who evaluated the thickness of the diaphragmatic muscle in six different areas, also revealed that the thickness of the left posterior pillar decreas-es with mechanical ventilation duration[23].

Considering that at moment there is no recommended “reference” points  for diaphragmatic measurements, the celiac axis was used as a reference point on both axial and coronal images based on previously published data[21]. However, recent data suggest, that multiple  reference points should be used considering the heterogenous structure of the diaphragm[23].

2) Reference was added and these 2 paragraphs: 

Moreover, it has also been demonstrated that diaphragm thickness correlates with skeletal muscle index in mechanical ventilated patients, including multiple trauma patients[23], [24].

Taking into account that skeletal muscle index is considered an independent risk factor for prolonged mechanical ventilation, our current research lacks further data regarding body composition.

3) "Between the CT exams, all patients received only pressure controlled mechanical ventilation" was reformulated.

4) "No patient received muscle relaxants during the analyzed period." was added.

Round 2

Reviewer 2 Report

Dear authors thank you for your answers.

Paragraph 2.1 Statistical Analysis: Please, indicate and specify what is "r" as reported, for example in lines 193/194. Please, indicate and specify what is "F" as reported in lines 173/174.

Line 106: you report "For the comparison of the two sets of measurements (on admission vs after day 5) Student t test for two independent groups was used." The appropriate test should be the Paired T test because the exams were done on the same patients. Please remake the analysis reporting the new results in whole the document. Furthermore I've some doubts if it is redundant reporting the t-test results when it was done the regression analysis, a better way to analyze this kind of data. Paired T-test does not consider the different follow-ups of the patients. So I would remove these kind of analysis.

Line 173 it is reported axial slope (F=7.52, p=0.094) as significant, but the p-value is 0.094. Can you check and correct, the p-value or the phrase, please?

Line 184/187, report CI 95% too, please. In general report CI 95% when possible.

Line 193, 194: Can you indicate what "r" is ?

Figure 8: could you adding the estimates of Spearman's correlation in the four panels of the figure?

Thank you

Author Response

Dear Reviewer,

I am grateful for your suggestion and the time you took to provide it.

I have made all the necessary changes, as follows:

Paragraph 2.1 Statistical Analysis: Please, indicate and specify what is "r" as reported, for example in lines 193/194. Please, indicate and specify what is "F" as reported in lines 173/174.

An explantion for the F-test was added in the text.

In the linear regression analysis, F-test derived from the ratio of mean square regression and  mean square residual, was used to evaluate if the variability  of the regression model is explained by the variation in the dependent variable or can be attributed to a random chance. In other words, a significant F-test validate the relationship between independent and dependent variables.

An explanation for correlation coefficient was also included:

Spearman correlation coefficient (r) was also measured in order to describe the rela-tions between the diaphragmatic thickness and other variables. Knowing that the corre-lation coefficient r ranges between -1 and 1, a negative r value indicates an inversely proportional relationship between variables, while a positive correlation confirms a proportional relation. When the r value is null, no correlation is validated

Line 106: you report "For the comparison of the two sets of measurements (on admission vs after day 5) Student t test for two independent groups was used." The appropriate test should be the Paired T test because the exams were done on the same patients. Please remake the analysis reporting the new results in whole the document. Furthermore I've some doubts if it is redundant reporting the t-test results when it was done the regression analysis, a better way to analyze this kind of data. Paired T-test does not consider the different follow-ups of the patients. So I would remove these kind of analysis. - paired t-Test analysis was removed.

Line 173 it is reported axial slope (F=7.52, p=0.094) as significant, but the p-value is 0.094. Can you check and correct, the p-value or the phrase, please?

Coronal slope (F=6.18, p=0.017) differs significantly from the value of 0 in statistical terms. The axial slope (F=7.52, p=0.094) did not meet the significance threshold

Line 184/187, report CI 95% too, please. In general report CI 95% when possible. – Not applicable because as you suggested we removed paired t-Test analysis.

Line 193, 194: Can you indicate what "r" is ? – explained in paragraph 2.1

Figure 8: could you adding the estimates of Spearman's correlation in the four panels of the figure?

R and P values were added in the figure.

Reviewer 3 Report

I have no further comments for the authors. I appreciate their responses.

Author Response

Thank you again for feedback !